# Comparative Evaluation of Dental Enamel Microhardness Following Various Methods of Interproximal Reduction: A Vickers Hardness Tester Investigation

**DOI:** 10.3390/biomedicines12051132

**Published:** 2024-05-20

**Authors:** Dan-Cosmin Serbanoiu, Aurel-Claudiu Vartolomei, Dana-Valentina Ghiga, Silvia Izabella Pop, Irinel Panainte, Marioara Moldovan, Codruta Sarosi, Ioan Petean, Marie-Jose Boileau, Mariana Pacurar

**Affiliations:** 1Faculty of Dental Medicine, GEP University of Medicine Pharmacy, Science and Technology of Targu Mures, 540139 Târgu Mures, Romania; serbanoiu.dancosmin@gmail.com (D.-C.S.); claudiu.vartolomei@gmail.com (A.-C.V.); valentinaghiga@gmail.com (D.-V.G.); irinel.panainte@yahoo.com (I.P.); marianapac@yahoo.com (M.P.); 2Raluca Ripan Chemistry Research Institute, Babes-Bolyai University, 400294 Cluj-Napoca, Romania; marioara.moldovan@ubbcluj.ro (M.M.); codruta.sarosi@ubbcluj.ro (C.S.); 3Faculty of Chemistry and Chemical Engineering, Babes-Bolyai University, 400084 Cluj-Napoca, Romania; ioan.petean@ubbcluj.ro; 4Faculty of Dental Medicine, Bordeaux University, CEDEX, 33076 Bordeaux, France; mariejoseboileau@gmail.com

**Keywords:** interproximal reduction, enamel microhardness, Vickers hardness tester, dental enamel

## Abstract

Interproximal enamel reduction, also known as stripping, is a common orthodontic procedure that reduces the mesiodistal diameter of teeth, allowing for a balance of available space in dental arches. The aim of this study was to assess the enamel surface microhardness resulting from the application of currently available methods for interproximal reduction. Forty-two extracted human permanent teeth were divided into six different groups, each subjected to a therapeutic stripping procedure using various methods (i.e., diamond burs, abrasive strips of 90 μm, 60 μm, 40 μm, and 15 μm, and abrasive discs). Stripping was performed by a single individual in accordance with the manufacturers’ recommendations for the various systems used. One of the proximal faces of the tooth underwent IPR, while the other side remained untreated for control. The hardness of the enamel surface was measured using a Vickers hardness tester. The control group achieved the hardest enamel surface (354.4 ± 41.02 HV1), while the lowest was observed for enamel surfaces treated with 90 µm abrasive strips (213.7 ± 118.6). The only statistically significant difference was identified in comparisons between the values measured for the control group and those obtained after stripping with diamond burs (*p* = 0.0159). Enamel microhardness varied depending on the stripping instrument used, but no statistically significant differences were found (*p* > 0.05). Optimal microhardness values, close to those of healthy enamel, were achieved after mechanical treatment with 15 µm abrasive strips and abrasive discs. Dental stripping is a safe therapeutic procedure that has a relatively minor influence on the microhardness of surface enamel.

## 1. Introduction

Interproximal enamel reduction (IER), also known as enamel stripping, constitutes a methodical approach to diminishing the mesiodistal dimension of teeth. This technique facilitates the harmonization of available arch space with the quantity of dental substance (width of the teeth). The quantum of space liberated through this method can oscillate from 8–10 mm (in exceptional scenarios) to a commonly achieved range of approximately 5 mm, thereby enabling the amelioration of dental incongruities characterized by mild-to-moderate crowding [1,2,3]. The adoption of dental stripping (IER) is increasingly favored within the realm of orthodontic therapeutics, especially in conjunction with the use of removable orthodontic aligners [4,5].

Hence, this procedure serves as a viable substitute for extraction therapies, attributing its potential to notable decreases in the duration of treatment while preserving the transverse dimension of the dental arch and the anterior inclination of the lower incisors [6,7]. The principal indications for implementing the interproximal enamel reduction (IER) technique include the correction of tooth size discrepancies across the maxillary and mandibular arches as denoted by the Bolton Index, management of mild-to-moderate dental crowding, normalization of gingival contours, adjustment of the Spee curve, augmentation of post-orthodontic treatment stability to prevent relapse, and the recontouring of proximal contacts to eliminate interdental “black triangles” arising from gingival recession [8,9]. Additionally, IER proves beneficial in dental reshaping scenarios, such as in the esthetic modification of canines into lateral incisors in cases of congenitally absent lateral incisors following space closure. Moreover, in mixed dentition phases, IER may be strategically employed in managing moderate crowding among growing patients or during functional therapies to reallocate space in anticipation of the eruption of permanent dentition or when the deciduous molar must be retained because of a congenitally missing succedaneous premolar [2,10,11,12].

The methodology behind this therapeutic intervention has undergone substantial refinement in recent years. Various techniques have been evaluated to establish an optimally effective, efficient, and suitable protocol to facilitate the stripping process in a manner that not only secures the necessary space for achieving therapeutic objectives but also ensures the attainment of impeccably smooth dental surfaces [9].

The body of literature concerning enamel reduction exhibits considerable variability, leading to divergent reports on the quantifiable extent of enamel that can be safely removed, as articulated by various authors within the field. Initially, the proposition was made that interproximal enamel reduction (IPR) of the lower incisors could yield up to 3 mm of interdental space [13]. Further analyses posited that in excess of 6 mm could be garnered through IPR applied to premolars and molars, contingent upon a reduction of 0.4 mm at each proximal contact [14]. John Sheridan has advanced the notion that the removal of 50% of the total extant proximal enamel across the dental arch can facilitate the acquisition of 8.9 mm of space. Of this, 6.4 mm emanates from the eight tooth-to-tooth contact points located in the lateral segment extending from the first premolar to the second molar. Conversely, 2.5 mm of space is engendered via stripping at the five contact points situated in the anterior region [14,15].

It is imperative for the clinician to conduct an accurate diagnostic assessment to ascertain the exact volume of enamel that necessitates removal, thereby circumventing excessive reduction and, consequently, the onset of iatrogenic complications [16]. In this context, a careful clinical evaluation of the integrity of dental surfaces will be conducted, correlated with the patient’s oral hygiene status and chronological age, which guide us regarding the position of the pulp chamber and implicitly the amount of enamel that can be safely removed. Dental morphology shows us that not all teeth are candidates for interproximal enamel reduction. Teeth with a triangular shape are much better candidates for the stripping procedure compared to cylindrical or rectangular ones [17]. For a better appreciation of dental morphology, orthodontic practitioners can use the Le Huche index (mesiodistal width at the contact point/mesiodistal width at the cervical margin), which highlights the varying degrees of divergence of dental crowns. The higher the index, the more it can be considered that the tooth has a triangular shape and, according to Langlade, the more suitable it is for the stripping procedure [17].

Additionally, based on radiological investigations included in the orthodontic records (panoramic radiography, Cone-Beam Computed Tomography—CBCT), we can more precisely evaluate the morphology and thickness of hard dental tissues, thereby ensuring the amount of enamel removed through interproximal reduction (IPR) [18].

Excessive removal of enamel can compromise a tooth’s morphology and underlying structure, reducing its strength and exposing it to cracks and dental fractures.

Dental enamel, a mineralized tissue of epithelial origin, serves as the protective outer layer of the tooth crown and is recognized as the hardest biological tissue in the human body. Its composition is predominantly inorganic, consisting of 92–96% mineral content, with the remainder comprising approximately 3% water and 1% organic constituents by weight [19,20].

Microhardness stands as the paramount metric for detecting changes in the mechanical characteristics of mineralized tissues and serves as an efficacious predictor for other critical mechanical parameters, including Young’s modulus and yield strength [21].

The traditional evaluation of microhardness in dental science is conducted utilizing either Vickers or Knoop hardness tests. The Knoop indenter is capable of detecting deformations extending to 20 μm in length, whereas the Vickers indenter permits the measurement of areas as diminutive as 5 μm in diameter, indicating a higher sensitivity of the Vickers indenter towards plastic deformations [22].

Surface microhardness is a physical property of enamel and dentin surfaces, correlating with the mineral composition of the dental structure [23]. In healthy dental enamel, the hardness value decreases progressively from the surface towards the amelo-dentinal junction (from the exterior towards the interior of the tooth), due to a reduction in mineral content and density. The calcium and phosphate levels within the structural component of enamel diminish towards dentin. Thus, it can be clearly considered that the microhardness of the dental surface is closely dependent on the mineral content of calcium and phosphate within the enamel [24,25,26].

Thus, based on this observation, the excessive removal of surface enamel can influence and jeopardize proper dental stability. Although numerous studies focus on the surface irregularities resulting from grinding and polishing procedures, very few studies concentrate on the impact these two operations can have on the surface microhardness of enamel and their effects on dental strength.

The aim of this study was to evaluate the effect of commonly used interproximal reduction (IPR) instruments on the microhardness of enamel using Vickers microhardness tests. The null hypothesis tested was that there is no relationship between enamel microhardness and the various methods of interproximal reduction.

## 2. Materials and Methods

This study was conducted on a total of 40 extracted human permanent teeth obtained from the Oral and Maxillofacial Surgery Department of the George Emil Palade University of Medicine, Pharmacy, Science, and Technology of Târgu Mureș, with the approval of the Ethics Committee (No. 2157 from 20 March 2023).

Teeth exhibiting fillings, scale, staining, demineralization, advanced caries, fluorosis, enamel cracks, or defects, as well as prosthetic restorations on the mesial or distal surfaces, were excluded from this study. The selected teeth were meticulously cleansed under a stream of water, with all blood, adhering soft tissues, and calculus being eliminated through delicate mechanical debridement using a toothbrush of medium stiffness.

Following debridement, the teeth underwent disinfection through immersion in a 1.0% chloramine-T trihydrate solution (pH 9.1) with bacteriostatic and bactericidal properties for a period not exceeding one week, in alignment with the specifications of ISO 3696:1987 (International Organization for Standardization ISO/TS 11405, third edition, 1 February 2015). To replicate intraoral conditions, the specimens were thereafter preserved in an artificial saliva medium. To prevent degradation, the storage solution was refreshed at a minimum bi-monthly interval. The composition of the artificial saliva included disodium phosphate, sodium bicarbonate, calcium chloride, hydrochloric acid, and distilled water, with specific concentrations detailed in Table 1 [27], and it was prepared at the Raluca Ripan Institute for Research in Chemistry in Cluj Napoca, adhering to findings identified in other research studies [28].

To emulate in situ conditions, the teeth were anchored within a fixed plaster model. A condensation silicone material (Optosil; Heraeus Kulzer, Hanau, Germany) was employed to secure the teeth, facilitating a semblance of natural tooth mobility within the plaster framework, as opposed to direct encasement. Enamel stripping procedures were uniformly performed by the same practitioner, adhering to the instrument manufacturers’ guidelines. For operations using burs, handpieces were utilized at a velocity of 400,000 rpm with water cooling; conversely, for discs and abrasive strips, handpieces were adjusted to a speed of 5000 rpm.

Enamel reduction was performed using 8392 “mosquito” fine grit-stripping burs, delineated by a red color code, size 0.16, produced by Komet, USA. In terms of stripping strips, all four gradations—15 microns, 40 microns, 60 microns, and 90 microns—were employed, manipulated via a stripping handpiece from Task Inc, operating at a velocity of 5000 rpm with oscillatory actions of 1.3 mm. For abrasive interventions, Sof Lex polishing discs (fine), fabricated by 3M Deutschland and identified by an orange color code for abrasiveness, were utilized.

For each enamel specimen, a distinct IPR instrument (i.e., bur, strip, and disc) was utilized once before replacement to maintain instrument integrity. To achieve uniform enamel reduction across all specimens, interproximal reduction (IPR) was carried out over the entire thickness of the stripping instrument (up to 0.5 mm), which was activated against the enamel surface for a duration of 30 s. This study divided 40 teeth into 6 cohorts, each aligned with a specific stripping instrument, as delineated in Table 2. Consequently, every group comprised 7 samples, with the enamel reduction procedure executed on either the mesial or distal aspect of the tooth. Conversely, the enamel on the contralateral side served as an untreated control to facilitate comparative analysis.

Upon concluding the IPR procedures, the specimens were subjected to cleansing with a 0.9% sodium chloride solution and subsequently dried utilizing the air spray feature from the dental unit. Thereafter, they were placed and preserved in artificial saliva until surface microhardness measurements of the dental enamel were performed.

Mechanically processed dental specimens subjected to the action of various stripping devices were presented for investigation, as follows: burs; IPR strips with abrasive particles of diameters 90, 60, 40, and 15 μm; and polishing abrasive discs. The procedure involved slicing thin enamel sections from the processed area for detailed investigations of enamel microhardness, as well as a section from the unprocessed part for control purposes.

The hardness of the enamel surface was measured using a Vickers hardness tester, Duramin-40 AC3, produced by Struers Company, Ballerup, Denmark. HV1 hardness values were recorded at an indentation time of 8 s. The Vickers hardness number (VHN) was calculated for each enamel surface using the formula: $$VHN=1854.4\times \frac{F}{d^{2}},$$
where *F* = load (g) and *d* = average of indentation diagonals (μm). A minimum of 3 indentations were performed on each specimen to establish the average value. The determined values were averaged to represent the Vickers hardness value of that specimen.

Statistical evaluation included descriptive statistics (mean, median, standard deviation) and inferential statistics. The Shapiro–Wilk test was utilized to assess the distribution of the data series. For comparisons of means, parametric Student’s *t*-test was applied, and for median comparisons, the non-parametric Mann–Whitney test was conducted. A significance level of 0.05 was selected for the *p*-value. All statistical analyses were performed using the trial version 10 of GraphPad Prism software, Dotmatics, Boston, MA, USA.

## 3. Results

### 3.1. Vickers Hardness Tester Investigation

The HV1 indentation imprints highlighted by the optical system of the hardness tester revealed alterations to the enamel surface following the grinding process as well as following the action of 90 μm abrasive strips. Under these circumstances, it is anticipated that the morphological changes to the surface would affect the hardness of the enamel. All specimens were tested with the Vickers hardness tester in accordance with the HV1 trial, with the obtained imprints presented in Figure 1. The unit of measurement for hardness obtained through this test is known as the Vickers pyramid number (HV). The hardness number can be converted into SI units (MPa or GPa) by multiplying the HV result by 9.807 for MPa and by 0.009807 for GPa.

It was found that the average value for the control sample (i.e., untreated healthy enamel) was 354.40 HV1 (3.475 GPa), serving as the baseline reference for the variation chart in Figure 2 and fully in agreement with data from the specialized literature [29,30]. The grinding treatment resulted in an average hardness of 225.83 HV1 (2.214 GPa), while the use of 90 µm abrasive strips leads to a mean value of 213.71 HV1 (2.095 GPa). The use of diamond burs and 90 µm abrasive strips generated an intense degradation effect at the enamel surface, as evidenced by images provided by the optical system of the Vickers hardness tester, which explain the reduced microhardness values (Figure 2).

Consequently, this drastic decrease in hardness values confirms the morphostructural observations and correlates with the induction of increased roughness values, a fact also highlighted in the specialized literature [30]. Strips with 60 µm abrasive material also resulted in a reduced average hardness value of 264.00 HV1 (2.589 GPa), but with a slight increase compared to the hardness achieved after treatment with the 90 µm strips. This upward trend in average hardness values progressively continues with the observation of morphostructural improvements, such that after treatment with 40 µm abrasive strips, values closer to healthy enamel are obtained, namely 284.66 HV1 (2.791 GPa). The exceptional quality of the surface following the smoothing effects of the 15 µm abrasive strips and polishing discs leads to slightly higher hardness values of 314.21 HV1 (3.081 GPa) and 291.50 HV1 (2.858 GPa), respectively.

### 3.2. Statistical Analysis

The results of the Vickers microhardness test are presented in Table 3. The hardest enamel surface was obtained for the control group, with a hardness of 354.44 ± 41.02 HV1. On the opposite end, the lowest microhardness was recorded after mechanical treatment with 90 µm abrasive strips, although no statistically significant differences (*p* = 0.1508) were observed in the comparisons between the two groups.

The only statistically significant difference (*p* = 0.0159) was identified in comparisons between the values measured for the control group and those obtained after mechanical treatment with diamond stripping burs (225.8 ± 43.49), as can also be seen in Table 3.

Regarding the remaining comparisons, no statistically significant differences (*p* > 0.05) were identified between the enamel microhardness values recorded in the control group and those obtained following the application of the various stripping instruments used in our study (Table 3). Thus, the null hypothesis with which this study commenced was accepted, implying that there is no correlation between the microhardness of dental enamel and different methods of interproximal enamel reduction.

Furthermore, when comparing the values resulting from mechanical treatment with the various stripping instruments used in the current study, no statistically significant differences (*p* > 0.05) were identified between the values recorded for enamel microhardness across any of the systems (Table 4).

## 4. Discussion

Dental crowding represents the predominant characteristic encountered within orthodontic patients. Various non-extraction strategies are employed to address this issue, including expansion of the dental arch, proclination of the anterior teeth, distal movement of dentition, derotation of the posterior rotated teeth, uprighting of tilted molars, and interproximal enamel reduction [31]. Interproximal enamel reduction is a frequently employed orthodontic procedure for correcting dento-maxillary disharmonies with crowding, both primary, due to tooth size discrepancies, and secondary to other functional or hereditary causes of the patient [8,10,32,33]. This therapeutic technique has gained significant advancement over the last decade with the refinement of stripping techniques, particularly in conjunction with orthodontic therapy using aligners [5,34]. 

Beyond functional corrections, dental stripping also enhances esthetic appeal and smile harmony by reducing black triangles between adjacent teeth, often caused by gingival retractions [8]. 

However, like any medical procedure, dental stripping presents a series of disadvantages related to a reduction in enamel quantity on the tooth surface; thus, many dental professionals are convinced that this application leads to dental sensitivity, dental fissures, reductions in enamel hardness, caries, and ultimately impacts tooth stability on the arch [2,6,35,36]. Although a direct relationship between stripping procedures and an increased susceptibility to caries and periodontal diseases cannot be established, certain preventive strategies have been recommended following enamel reduction to mitigate potential adverse effects [2,3,37].

Given the numerous advantages outlined in the dental literature and the currently widespread use of the interproximal enamel reduction (IER) method, this study aimed to quantify its potential negative side effects on the microhardness of dental enamel and to clarify any uncertainties related to IER among orthodontic clinicians. Multiple research endeavors have also explored the detrimental effects associated with enamel stripping [38]. 

Hardness is commonly defined as a material’s resistance to deformation. However, hardness is not an intrinsic property, as the hardness value of a material results from a specific measurement procedure [39]. Vickers microhardness is a well-established and suitable method for investigating the properties of human hard dental tissues as well as dental restorative materials [25,39,40,41,42,43]. Therefore, our study focused on evaluating the microhardness of dental enamel resulting from various stripping methods using the Vickers hardness tester, rather than the Knoop hardness tester.

Our study demonstrated a significant decrease in enamel microhardness through grinding using diamond IPR burs or the use of 90 µm abrasive strips. In contrast, the mean enamel microhardness values in the groups where 15 µm abrasive strips and Sof-Lex abrasive discs were used for stripping were similar to those in the control group. This finding suggests that the use of 15 µm abrasive strips and Sof-Lex discs preserves the hardness of intact enamel surfaces. Therefore, it can be stated that these methods do not cause additional grinding of the enamel structure and can be safely used. Similar results were observed in a study conducted by Arman and colleagues, where the rationale behind the microhardness evaluations of stripped enamel surfaces was that enamel microhardness is directly related to the mineral content of enamel, and the mean enamel microhardness values in samples of permanent teeth did not show significant differences between the groups they studied [11,44]. 

Thus, dental stripping, being a mechanical and not a chemical medical procedure, interferes with and influences the mineral component of enamel and, consequently, the hardness of the tooth to a lesser extent.

An interesting observation presented in a previous study is that intact enamel naturally has a more irregular surface than enamel subjected to dental stripping and mechanical polishing [45]. Although bacterial plaque accumulation and subsequent colonization are directly proportional to surface irregularities [46], intact enamel is more resistant to demineralization due to the presence of the outermost layer, which is less soluble and rich in fluoroapatite [47,48]. Previous in vitro investigations have shown that abraded enamel surfaces are more susceptible to demineralization than intact enamel surfaces. This has been attributed to the removal of the outermost aprismatic enamel layer, leading to a decrease in enamel microhardness [45]. Dental stripping, which removes a variable amount (0.2–0.5 mm) from the outer surface of the tooth, specifically from the aprismatic enamel, raises the question of whether it negatively affects the hardness of the enamel and the tooth’s resistance within the dental arch [49,50].

Unfortunately, studies that focus on evaluating the microhardness of enamel following therapeutic stripping procedures are relatively few. These studies tend to rely more on assessing the surface roughness of the enamel after interproximal reduction (IPR) and on evaluating the post-stripping caries risk [9,11,32,37,51]. 

An initial conclusion from reviewing articles on this topic indicates that the surface modifications induced in dental enamel by stripping vary depending on the technique and stripping instrument chosen. Abrasive strips and diamond burs with higher roughness produce significant enamel surface alterations by destabilizing the binding protein material that ensures cohesion between hydroxyapatite nanoparticles and through intense destructuring effects. While these methods ensure efficient grinding with high yield in terms of creating the necessary space for correcting dental crowding, they leave behind increased roughness unsuitable for tooth health and reduced hardness [45]. The use of 15 µm abrasive strips and polishing abrasive discs in the stripping procedure safely removes dental enamel, achieving a surface similar to that of intact enamel and very good microhardness values [10,37,45,47,49,52].

Therefore, as a clinical recommendation, we can discuss the finishing and polishing of dental surfaces subjected to stripping with diamond burs and rough abrasive strips, subsequently using 15 µm strips and abrasive polishing discs to ensure optimal hardness and smoothness of the enamel.

The rationale behind our research was based on the fact that enamel hardness is closely linked to the presence of the outer aprismatic enamel layer, which undergoes changes following dental stripping. Although our study sheds light on this topic, contributing to the clarification of many ambiguities and differentiating the effects of various stripping methods on enamel microhardness, there are also several inherent limitations related to our method of evaluating microhardness, which cannot use unprocessed enamel surfaces, thus altering the mineral-rich outer enamel layer and, therefore, differing from the clinical situation. Preservation of the outer enamel layer more accurately reflects the clinical context and can be quantified through techniques of superimposition and subtraction [53].

Additionally, enamel microhardness is closely related to its inorganic phospho-calcium component and the ongoing process of enamel remineralization in the oral cavity. As the remineralization of an enamel surface in the oral cavity is a complex process, the limitations of this in vitro study include the difficulty of precisely simulating biological aspects; artificial saliva is composed only of inorganic ions, and the effects of salivary proteins, pellicle, and plaque on mineralization inhibition have not been considered [52]. Other confounding factors involve the possibility of experimental errors and differences in the enamel’s microstructure between samples. Additional clinical studies are necessary to authenticate the results for future clinical applications.

This study may assist in evaluating the real contribution of enamel hardness to wear phenomena, both naturally through enamel attrition and as a result of various dental therapeutic procedures, in future works and studies. Additionally, all biological responses remained unaddressed in this study, particularly the reactions of the pulp–dentin complex during and after therapeutic stripping procedures. The effects generated by the force/pressure applied to the tooth during IPR and the temperature induced by the action of various stripping instruments are planned for future ex vivo and in vivo investigations. Moreover, we aim to research the microhardness of enamel subjected to dental stripping after the application of remineralization products used in dental practice in subsequent studies.

## 5. Conclusions

The use of the Vickers hardness tester for evaluating enamel microhardness following various methods of interproximal enamel reduction demonstrated differing values depending on the instrument used for stripping. Specifically, diamond burs and 90 µm abrasive strips exhibited increased efficiency in the removal of approximal enamel, but negatively impacted enamel microhardness, significantly reducing it. Values close to those of healthy enamel in terms of microhardness were achieved after using 40 and 60 µm abrasive strips. There was a progressive trend of increasing average microhardness values, such that after using 15 µm abrasive strips and Sof Lex abrasive discs, the microhardness values obtained were appreciably higher, compared to those recorded for healthy enamel.

## Figures and Tables

**Figure 1 biomedicines-12-01132-f001:**
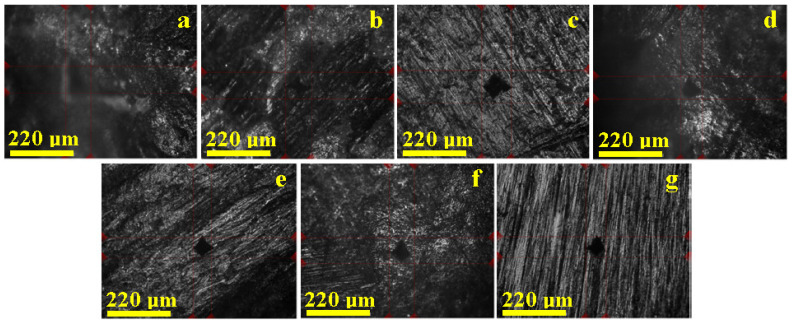
HV1 indentation imprints highlighted by the optical system of the hardness tester: (**a**) control sample—unground enamel, (**b**) burs, (**c**) 90 µm strips, (**d**) 60 µm strips, (**e**) 40 µm strips, (**f**) 15 µm strips, and (**g**) discs.

**Figure 2 biomedicines-12-01132-f002:**
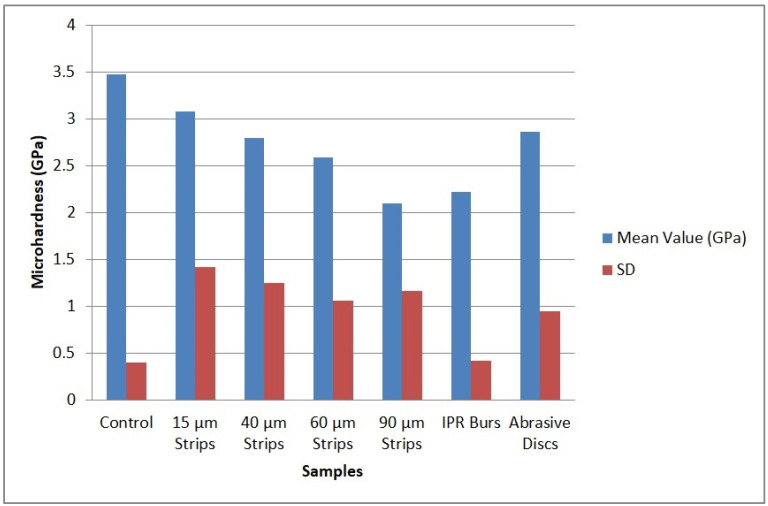
Variation in enamel microhardness according to the applied treatment.

**Table 1 biomedicines-12-01132-t001:** Artificial saliva components.

Composition	Percentage %
Na_2_HPO_4_ NaHCO_3_ CaCl_2_	0.3
HCl-1M	0.3
H_2_O	99.4

**Table 2 biomedicines-12-01132-t002:** IPR instruments used in this study.

IPR Instrument	Model	Manufacturer	Grit	Handpiece
Abrasive strips 15 microns	EVA active on both sides	Task Inc., Tokyo, Japan	15 μm	Slow speed (5000 rpm)
Abrasive strips 40 microns	EVA active on both sides	Task Inc., Tokyo, Japan	40 μm	Slow speed (5000 rpm)
Abrasive strips 60 microns	EVA active on both sides	Task Inc., Tokyo, Japan	60 μm	Slow speed (5000 rpm)
Abrasive strips 90 microns	EVA active on both sides	Task Inc., Tokyo, Japan	90 μm	Slow speed (5000 rpm)
Burs	8392 “mosquito” bur	Komet, Rock Hill, South Carolina, USA	Red Fine grit	High speed (400,000 rpm) with water cooling
Discs	Sof Lex System Kit	3M, Neuss, Germany	Orange Fine 20 μm	Slow speed (5000 rpm)

**Table 3 biomedicines-12-01132-t003:** Vickers microhardness value using different IPR instruments (HV1/8 s).

Instrument	Sample No.	Mean ± SD	Min	Max	*p* Value
15 μm abrasive strips *	7	314.2 ± 145	148.1	499.6	0.8329 ^†^
40 μm abrasive strips *	7	284.6 ± 127.5	143	457.1	0.2677 ^†^
60 μm abrasive strips *	7	264 ± 108.4	143.1	420.3	0.2222 ^†^
90 μm abrasive strips *	7	213.7 ± 118.6	149.3	423.9	0.1508 ^†^
Diamond burs *	7	225.8 ± 43.49	184.2	285.3	0.0159
Abrasive discs *	7	291.5 ± 96.04	148	395.8	0.4286 ^†^
Control	7	354.4 ± 41.02	313.1	419.3	-

* Relative to control; ^†^ Indicates nonsignificant *p* value (*p* > 0.05).

**Table 4 biomedicines-12-01132-t004:** Multiple comparisons of Vickers microhardness created by IPR instruments (*p* values).

Instrument	15 μm Abrasive Strips	40 μm Abrasive Strips	60 μm Abrasive Strips	90 μm Abrasive Strips	Diamond Burs	Abrasive Discs
15 μm abrasive strips	-	0.6943 ^†^	0.7242 ^†^	0.2222 ^†^	0.5697 ^†^	0.6620 ^†^
40 μm abrasive strips	0.6943 ^†^	-	0.8763 ^†^	0.2677 ^†^	0.4038 ^†^	0.8357 ^†^
60 μm abrasive strips	0.7242 ^†^	0.8763 ^†^	-	0.5476 ^†^	0.7302 ^†^	0.6623 ^†^
90 μm abrasive strips	0.2222 ^†^	0.2677 ^†^	0.5476 ^†^	-	0.2857 ^†^	0.4286 ^†^
Diamond burs	0.5697 ^†^	0.4038 ^†^	0.7302 ^†^	0.2857 ^†^	-	0.3524 ^†^
Abrasive discs	0.6620 ^†^	0.8357 ^†^	0.6623 ^†^	0.4286 ^†^	0.3524 ^†^	-

^†^ Indicates nonsignificant *p* value (*p* > 0.05).

## Data Availability

Data are contained within the article.

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
