# Peer review of "Comparative Evaluation of Dental Enamel Microhardness Following Various Methods of Interproximal Reduction: A Vickers Hardness Tester Investigation"

_biomedicines, 2024, doi:10.3390/biomedicines12051132_

Round 1
Reviewer 1 Report
Comments and Suggestions for Authors interesting original study in design, explanation and clinical application Comments on the Quality of English Language good fluent and understandable languageAuthor Response
Dear Sir/Madam,
Thank you very much for the review given and I appreciate the interest shown. The topic addressed in the article is very technical and I tried to use an appropriate language that includes specialized terms and matching semantics. My desire was to respect the English and structure present in the specialized literature and in articles that similarly address the same topic. Once again, thank you!
Reviewer 2 Report
Comments and Suggestions for Authors
Interesting article, because the topic of enamel protection is extremely important in dentistry. Therefore, every practical publication in this field is extremely valuable for a dentist. As a recnent, I have a few comments:
Introduction:
Line 79
John Sheridan has advanced the notion that the…..[15]- in ref 14. Sheridan JJ. The physiologic rationale for air-rotor stripping. J Clin Orthod. 1997 Sep;31(9):609–12, please correct it.
Line 85
It is imperative for the clinician to conduct an accurate diagnostic assessment to ascertain the exact volume of enamel that necessitates removal- It would be good to add 1-2 sentences - what are the diagnostic methods to determine safe enamel reduction?
Line 108
Therefore, enamel microhardness could serve as an indirect indicator of the calcium and phosphate content within the enamel- This is not an analytical method for determining the Ca and PO4 content. Additionally, as you know, hardness will vary individually. I would suggest that you formulate this idea differently
Materials and methods
You are testing microhardness, it would be good to add a literature reference to this method
Line 131
in alignment with the specifications of ISO 3696:1987.- this also requires some explanation as to why you refer to this standard. Because this ISO standard applies to laboratory water. Add it as Ref,
Line 134
The composition of the artificial saliva included disodium phosphate, sodium bicarbonate, calcium chloride, hydrochloric acid, and distilled water, with specific concentrations detailed in Table 1-
This saliva composition, please add the ref? For example: Alhotan, A.; Raszewski, Z.; Alamoush, R.A.; Chojnacka, K.; Mikulewicz, M.; Haider, J. Influence of Storing Composite Filling Materials in a Low-pH Artificial Saliva on Their Mechanical Properties—An In Vitro Study. J. Funct. Biomater. 2023, 14, 328. https://doi.org/10.3390/jfb14060328
Or Kanathe, Pooja & Jain, Ruchi & Jain, Nilesh & Jain, Surendra. (2021). Formulation and Evaluation of Orodispersible Tablet of Fluvastatin Sodium. Journal of Drug Delivery and Therapeutics. 11. 42-47. 10.22270/jddt.v11i1.4498.
It would be good to post a photo of your jaw with your teeth embedded in Optosil plus plaster. It would make it easier for others to deal with a similar ex in the future.
For each manufacturer, please add the city and, if it is from the USA, an additional state.
Line 189
GraphPad Prism software.- producer city, country, please add
Results
Figure 2, please add the units on the Y axis, If you made 3 measurements, there was some SD, which would be a good idea to add to Figure 2
The corresponding units of HV are then kilogram-force per square millimetre (kgf/mm²). To convert a Vickers hardness number in SI units (MPa or GPa) one needs to convert the force applied from kgf to newtons and the area from mm2 to m2 to give results in pascals (1 kgf/mm² = 9.80665×106 Pa
https://www.wikidoc.org/index.php/Vickers_hardness_test
line 243
) were identified between the values recorded for enamel microhardness 243across any of the systems (Tabel 4).- Table 4, should be in the text after the description in the text.
Discussion
I like it!
References
Dorobat V, Stanciu D. Ortodontie si ortopedie dento-faciala. Bucuresti: Editura Medicala; 2014.- please add the pages.
Good luck in your further research!
Comments on the Quality of English Language
small improvement needed
Author Response
Thank you very much for your interest and the reviews provided. I have implemented the corrections suggested by you. I would like to add a few notes to facilitate communication. Unfortunately, we did not capture a proper photograph during the mechanical stripping procedure with the teeth embedded in the plastic model. Thank you once again for your help!
Reviewer 3 Report
Comments and Suggestions for Authors
1. Reduce the length of abstract.
2. Add some dental composite history in Introduction section and add recommended article into it.
A. Investigation of the physical, mechanical and thermal properties of nano and microsized particulate-filled dental composite material
B. Dynamic mechanical analysis of zinc oxide and hydroxyapatite particulate filled dental restorative composite materials
3. Add limitation of current study.
4. Add future scope of this study.
5. Figure 2: Why did 90 micrometer strips show low microhardness
6. Why did authors not perform other mechanical properties.
Author Response
Thank you very much for your interest and the reviews provided. I have implemented the corrections suggested by you. I would like to add a few notes to facilitate communication. Together with my colleagues, we conducted additional measurements beyond hardness, specifically regarding enamel roughness, which we studied using both an electron microscope and an atomic force microscope. These results are included in another study submitted for publication. In the current study, we aimed to focus solely on the effects of stripping procedures on enamel hardness and did not investigate the various composite dental materials used in dental practice. We certainly plan to consider this topic for future research. Thank you once again for your help!